# Evaluation of Theoretical Frameworks to Detect Correlates of HPV Vaccination in the Midwest, US, Using Structural Equation Modeling

**DOI:** 10.3390/vaccines11121856

**Published:** 2023-12-15

**Authors:** Abraham Degarege, Shinobu Watanabe-Galloway, Kristyne Mansilla, Rahel M. Sileshi, Edward S. Peters

**Affiliations:** Department of Epidemiology, College of Public Health, University of Nebraska Medical Center, Omaha, NE 68198, USA; swatanabe@unmc.edu (S.W.-G.); kmansilladubon@unmc.edu (K.M.); rsileshi@unmc.edu (R.M.S.); epeters@unmc.edu (E.S.P.)

**Keywords:** health theories, SEM, HPV vaccine uptake, US

## Abstract

Knowledge of a valid, well-designed, and targeted theory-based framework helps better characterize reasons for HPV vaccine hesitancy and identify promising approaches to increase vaccination rates for eligible individuals. This study evaluated health theories in explaining factors affecting HPV vaccination and used a theoretical framework to identify direct and indirect predictors and mediators of HPV vaccination. A cross-sectional survey regarding HPV vaccine uptake and related factors was conducted among 1306 teenagers and young adults in the Midwest, US, in March and April 2023. Structural equation modeling confirmed fit of the framework based on the Integrated Health Theory (IHT) to the HPV vaccine data (Comparative Fit Index = 0.93; Tucker-Lewis Index = 0.92; Root Mean Square Error of Approximation = 0.053). While willingness to uptake the HPV vaccine directly predicted increased uptake (*p* < 0.001), perceived benefits (*p* < 0.001) and barriers (*p* < 0.023) about the vaccine indirectly predicted increased and decreased uptake, respectively. In turn, beliefs about susceptibility (*p* = 0.005) and severity (*p* < 0.001) of HPV infection and associated cancers and barriers to vaccination in general (*p* < 0.001) indirectly predicted willingness to uptake the vaccine. In conclusion, IHT can be appropriate in examining predictors of HPV vaccine uptake in teenagers and young adults in the US, particularly in the Midwest.

## 1. Introduction

Over 47,199 cancer cases caused due to infection with human papillomavirus (HPV) occur each year in the United States [1]. Some cancers caused by HPV infection, including cervical cancer, can be prevented through screening, but screening guidelines are unavailable for other HPV-associated cancers [2]. The prevalence of cancers caused by HPV infection is disproportionately high in some states in the Midwest region [3,4,5,6,7]. 

In order to effectively control these HPV-related cancers, HPV vaccination has been utilized in the US since 2006 [8,9]. Currently, in the US, the HPV vaccine is recommended for males and females between the ages of 9 to 26 years and can be offered to selected individuals aged 27 to 45 years [10]. Although the HPV vaccines have been shown to be safe and effective at reducing the prevalence of HPV infection and related cancers [9,11], the increase in the rate of uptake among eligible individuals (ages 13 to 26 years) in the US particularly in some states of the Midwest region is slow, lowering its potential public health impact [12,13]. In 2021, the complete HPV vaccination rate (2 or 3 doses) among adolescents aged 13 to 15 was only 58.5%, far lower than the Healthy People 2030 goal of 80% [14,15]. The HPV vaccination completion rate in some states in the Midwest region is much lower than 50% in some ethnic and gender population groups [13]. Evidence based intervention is necessitated to reduce HPV vaccine hesitancy and improve HPV vaccine uptake among the target population in the US, particularly in the Midwest. 

Factors affecting HPV vaccination can be individual, interpersonal, community or societal levels and some could be direct predictors while others could be mediators or moderators [16,17,18]. For example, an individual factor such as intention to uptake HPV vaccine may directly affect HPV vaccination, while another individual factor such as beliefs about HPV infection may have an indirect effect. Also, a community-level factor such as area deprivation may play a moderating role in predicting HPV vaccination. A health theory-derived framework that follows an integrative approach and advanced statistical methods (e.g., structural equation model) is required to understand the determinants of HPV vaccination better. However, most studies have assumed these factors as direct predictors of HPV vaccine uptake and applied the traditional logistic regression model to identify correlates of HPV vaccination.

Moreover, data on the specific reasons for the low HPV vaccine uptake rate in the Midwest region of the US are limited. Most notably, little is known about the epidemiology of HPV vaccination hesitancy–refusal and delay, constituting an important intermediary step between adolescents’ and young adults’ vaccination intentions and vaccine uptake [19]. A clear understanding of factors influencing vaccine uptake in different settings could help develop specific, relevant, acceptable, and effective interventions and inform policy that better aligns with adolescents’ and young adults’ norms, experiences, and expectations. A well-designed and targeted theory-based intervention can effectively change the behavior of eligible individuals or their parents and health care providers who could affect their decision to uptake the vaccine. 

Theories such as Health Belief Theory (HBT), Theory of Planned Behavior (TPB), Theory of Reasoned Action (TRA), Social Cognitive Theory (SCT) and Socioecological Theory (SET) offer conceptual frameworks for understanding the dynamics of human behavioral change in specific contexts [20,21,22,23]. The HBT proposes that personal health behavior will be affected directly by individuals’ beliefs about a disease (perceived susceptibility to illness and perceived severity of illness), perceived benefits and barriers of taking a recommended health action, perceived control/self-efficacy, or confidence in one’s ability to perform an action and cues to action or stimulus to undertake behavior [21]. On the other hand, the TPB, which is an extension of the Theory of Reasoned Action, posits that attitude and subjective norms (views of others or perceived social pressure) about behavior and perceived behavioral control (the ease with which it can be performed) together affect the intentions about the behavior which could, in turn, influence the behavior [20]. Attitudes, in turn, are affected by behavioral beliefs, subjective norms are influenced by normative beliefs, and perceived behavioral control is triggered by control beliefs [20]. SCT posits that the individual, environment, and behavior interact with each other to form the basis for behavior [22]. The SET suggests that the individual, community, institution, and policy level factors interact with each other to affect behavior [23].

HBT, TPB, TRA, SCT and SET were developed in the 1970s or 1980s [20,21,22,23]. An advanced theory called IHT that includes all the constructs of HBT and TPB and the additional elements “background factors, knowledge and awareness” that could affect health behavior was developed in 2009 [20]. The IHT suggests that attitudes and norms directly affect the intention to practice a behavior, but the effect of belief factors on the intention to practice behavior is indirect through attitudes and norms [20]. In turn, sociodemographic, knowledge, and personality traits indirectly influence attitudes, subjective norms, and self-efficacy by affecting belief factors [20].

As determinants of vaccination are mainly on an individual level [24,25,26], HBT and TPB that suggest individual beliefs, attitudes, and norms as the main predictors of health behavior were most frequently used to examine predictors and design interventions to improve HPV vaccination uptake [26,27,28]. Indeed, several studies reported the constructs of HBT and TPB—perceived risk, perceived benefits of a behavior, perceived barriers to a behavior, norms related to behavior, cues to action, self-efficacy—as important predictors of HPV vaccination [24,25,26]. A study has also reported a good fit of the IHT-derived framework to explain factors associated with willingness to uptake the HPV vaccine [29]. 

This study’s main objective was to identify a health theory that can better explain factors affecting HPV vaccine uptake and apply the theory to identify direct and indirect predictors of HPV vaccination. As IHT encompasses all the constructs of HBT and TPB plus background factors and knowledge and awareness variables, we hypothesized that the IHT fits the data on predictors of HPV vaccination better than the HBT and TPB. The rationale for this study is that knowledge of a valid health theory framework that explains that HPV vaccine uptake data will help better characterize reasons for HPV vaccine hesitancy and identify promising approaches to address the barriers and improve vaccination rates for eligible individuals. 

## 2. Materials and Methods

### 2.1. Study Design and Participants

Data were collected by deploying a cross-sectional survey administered through Qualtrics online panel services, a commercial survey sampling and administration company in the US [30]. Qualtrics has over 90 million online samples or market research panelists used for corporate and academic research. The panelists completed a standardized set of questions to create their profile in Qualtrics. The profiles were used to choose eligible respondents at random for surveys. For this study, Qualtrics online samples used a combination of actively managed, double-opt-in market research panels to recruit participants. The study participants were existing pools of research panel samples of the Qualtrics aged 13 to 26 years old living in the Midwest area in the US (i.e., Illinois, Ohio, Michigan, Indiana, Missouri, Wisconsin, Minnesota, Kansas, Iowa, Nebraska, South Dakota, North Dakota) and who had agreed to be contacted for research studies. Sampling was performed based on gender (female ~ 50% and male ~ 50%) and age group (age in years 13–17 ~ 45%; age in years 18–22 ~ 40%; age in years 23–26 ~ 15%) from the 12 states. Qualifying screening questions presented at the beginning of the survey were used to select eligible participants. After screening eligible participants, demographic screening questions were used to ensure the demographic distribution of the sample was representative of the Midwest population. The inclusion of participants was completed when quotas for each demographic group were reached. 

A random sample of research panels in Qualtrics aged 13 to 26 years old living in the Midwest area in the US, who agreed to be contacted for research studies, received an invitation email to participate in the online survey. Individuals were invited to participate in the survey if they were preregistered for Qualtrics Panels and completed a baseline proprietary survey. The typical survey invitation through email was generally very simple and generic and informed the estimated completion time and incentives. It contained a hyperlink that directed the respondent to the study consent page and the survey instrument, which was completed through Qualtrics. Survey invitations through email did not include specific details about the questionnaire contents and were instead kept very general to avoid self-selection bias. Participants were informed that the survey was for research purposes, and confidentiality, risks, and benefits were described at the beginning of the survey. After they assented, participants were directed to the questionnaire; after completing it, they were compensated. Qualtrics staff monitored responses.

### 2.2. Questionnaire

The survey had 58 questions divided into 15 sections that systematically explore factors influencing HPV vaccination hesitancy and uptake. The questionnaire was designed based on measures of the IHT, TPB, and HBT components [20,21,22,23]. In addition, previous studies that reported factors affecting HPV vaccine uptake based on the US population were referred to while developing the questionnaire [31,32,33]. Most of the items were previously tested/used for measuring constructs of IHT in the Indian population [29]. Moreover, the items used to measure the construct susceptibility and severity were previously validated [34].

The questions in the survey were grouped into seven constructs (knowledge and awareness about HPV infection and cervical cancer, beliefs about susceptibility to HPV infection or cervical cancer, beliefs about the severity of HPV infection or cervical cancer, beliefs about benefits of vaccination, beliefs about barriers to vaccination, attitudes about benefits/facilitators of HPV vaccination and attitudes about barriers to HPV vaccination) according to IHT, TPB, and HBT (Table 1). In addition, there were questions used to assess the uptake of HPV vaccine (“What is your HPV vaccination status?”), willingness to receive the HPV vaccine (“for those who chose partially vaccinated; would you get the remaining dose sometime in the next 6 months?”), subjective norms about HPV vaccination (“I believe society expects me to get the HPV vaccine”), cues to receive HPV vaccine (“If I knew a woman with cervical cancer, I would be motivated to get to HPV vaccine”), normative beliefs about HPV vaccine (“I have a responsibility to get vaccinated for the protection of others”), self-efficacy (“I believe I can succeed/achieve getting the HPV vaccine even when things are tough”) and demographic status of the study participants (age, gender, educational status, ethnicity). Response to HPV vaccine uptake was recorded as unvaccinated, partially vaccinated (one dose); fully vaccinated (2 doses); fully vaccinated (3 doses); do not know. The response to willingness to uptake HPV vaccine was recorded as “Yes definitely”, “Yes probably”, “No probably not” and “No definitely not”.

### 2.3. Sample Size

As the conceptual framework that guided the analyses in this study encompassed direct and indirect predictors of HPV vaccination willingness or uptake and constructs that were not measured directly, SEM was used for analyzing the data. When compared to the conceptual frameworks based on TPB and HBT, the SEM based on IHT was expected to have the highest number of parameter estimates (*n* = 184) (39 factor loadings + 65 variances + 25 covariances + 55 structural paths) (Figure 1). The acceptable power cut-off value of the participant-to-parameter ratio for an SEM to accurately estimate the parameters is 7 [35]. Thus, we needed a minimum of 1288 participants to adequately test the validity of the conceptual frameworks based on the IHT, TPB, and HBT. Fortunately, Qualtrics collected and provided data for 1306 individuals. 

### 2.4. Ethical Consideration

The study obtained ethical approval from the Institutional Review Board (IRB) of the University of Nebraska Medical Center (IRB # 0696-22-EP). As this study has a very minimal risk on the participants, who were existing pools of research panel lists of a data processor company―Qualtrics, the IRB waived parental or guardian consent/permission for teenagers. Panelists were invited to participate through email and opted in by activating a link that directed them to the study consent page and survey instrument. The participants assented by simply checking a box after reading the consent sheet with details about this study’s purpose, confidentiality, risks, and benefits. Only those who assented completed the survey.

### 2.5. Data Analysis

Conceptual frameworks based on HBT (Figure 1A), TPB (Figure 1B), and IHT (Figure 1C) guided the analysis plan. Data were analyzed using Mplus version 8 [36]. First, a Cronbach’s alpha coefficient was calculated to evaluate the reliability/internal consistency of the items in measuring the constructs of the health theories (α ≥ 0.7 acceptable) [37,38]. Then, confirmatory factor analysis was used to assess the validity of the items (factor loading ≥ 0.4 acceptable) [39], the association between latent factors, and to check the fit of the measurement model to the data. SEM was used to evaluate the fit of the conceptual frameworks proposed based on the health theories to the HPV vaccination uptake data. Fit or validity of the conceptual frameworks was checked using chi-square (acceptable if *p* > 0.05) and other fit indices, including comparative fit index (acceptable if CFI > 0.90), Tucker–Lewis Index (acceptable if TLI > 0.90), and root mean square error of approximation (acceptable if RMSEA is <0.08) [40,41]. As the models based on the IHT, TPB and HBM were not nested, model comparison was performed using CFI, TLI and RMSEA. After identifying the health theory-based framework that best fits the observed data, SEM was applied to determine the direct and indirect predictors of HPV vaccination willingness and uptake. As the HPV vaccine uptake data were categorical (i.e., multivariate normal distribution does not exist), the variance–covariance matrix with the weighted least squares estimation method was used to estimate the parameters for direct and indirect effects [42]. Miss-fitting models were re-specified following the conceptual framework soundness and modification indices outputs from Mplus [43].

**Figure 1 vaccines-11-01856-f001:**
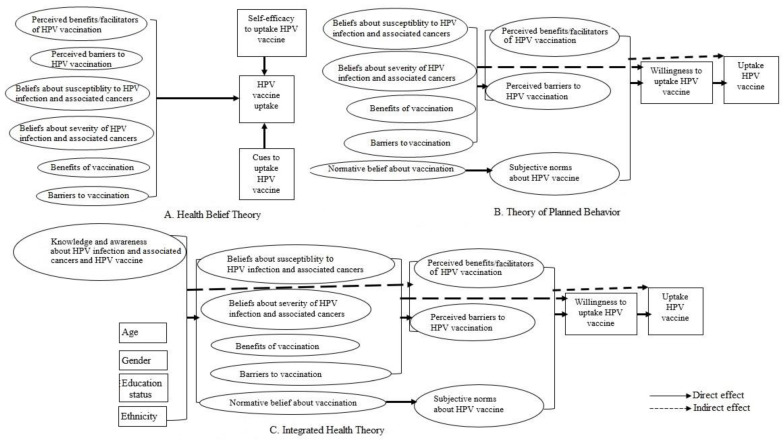
Proposed health theory-derived frameworks to understand predictors of HPV vaccine uptake among teenagers and young adults in the Midwest, US.

## 3. Results

A total of 1306 teenagers and young adults (age range: 13–26 years) participated in this study (Table 2). Close to 50% of the participants identified their gender as male and 45% as female, and 4.3% reported their gender as transgender or non-binary. The highest education level achieved by most participants was high school (73.3%). Only 26.6% reported college- or graduate-level education. Although the majority of the participants identified themselves as white (58.1%) and Black or African American (24.0%), American Indian or Alaska natives (5.3%) and Asians (3.1%) were also involved in this study. In terms of ethnicity, the majority were non-Hispanic or non-Latino (82.9%). Out of 1306 participants, 30% were fully vaccinated (2 or 3 doses), 9.5% were partially vaccinated (1 dose), 24.8% were unvaccinated and 35.3% did not know their vaccination status.

### 3.1. Measurement Models

#### 3.1.1. Integrated Health Theory

The measurement model for the IHT-derived conceptual framework involved the constructs “attitudes about the benefits/facilitators of HPV vaccination”, “attitudes about barriers to HPV vaccination”, “beliefs about the benefits of vaccination in general”, “beliefs about barriers to vaccination in general”, “beliefs about susceptibility to HPV infection and associated cancers”, “beliefs about severity of HPV infection and associated cancers” and “knowledge and awareness about HPV infection and associated cancers and HPV vaccine” (Figure 2). All the items in the original survey used for measuring these constructs loaded significantly (*p* < 0.001) with standardized factor loadings ranging from 0.54 to 0.84 for “attitudes about the benefits/facilitators of HPV vaccination” (Cronbach’s alpha, α = 0.83), from 0.50 to 0.84 for “attitudes about barriers to HPV vaccination” (α = 0.85), from 0.78 to 0.83 for “beliefs about the benefits of vaccination in general” (α = 0.86), from 0.40 to 0.80 for “beliefs about barriers to vaccination in general” (α = 0.70), from 0.74 to 0.92 for “beliefs about susceptibility to HPV infection and associated cancers” (α = 0.86), from 0.69 to 0.90 for “beliefs about severity of HPV infection and associated cancers” (α = 0.88) and from 0.53 to 0.69 for “knowledge and awareness about HPV infection and associated cancers and HPV vaccine” (α = 0.70). Data fit of this measurement model for the IHT-derived framework with all the items being free was fairly acceptable based on CFI (=0.918), TLI (=0.910) and RMSEA (=0.066, 95% CI 0.064, 0.068). However, the Chi-square test of model fit (x^2^ = 4311.23, *p* < 0.01) showed a lack of fit of the measurement model to the data. Thus, the measurement model was further modified by allowing covariance among residual terms of related constructs and the items in the same construct (Appendix A). This modification improved the model fit to the data (RMSEA = 0.054, 95% CI: 0.052, 0.056; CFI = 0.947, TLI = 0.940).

#### 3.1.2. Health Belief Theory and Theory of Planned Behavior

The measurement models for the HBT and TPB were identical and had all the constructs of the IHT except the variable “knowledge and awareness about HPV infection and related cancers and HPV vaccine” (Figure 2). The original measurement model that leaves all the items free did not fit the data (CFI = 0.735; TLI = 0.722; RMSEA = 0.098; χ^2^ = 13378.629, df = 987, *p* < 0.001). The measurement model remained unfit to the data (CFI = 0.751; TLI = 0.733; RMSEA = 0.096; χ^2^ = 12612.15, df = 967, *p* < 0.001), even after adding covariance among residual terms of related constructs and the items in the same construct (Appendix A). 

### 3.2. Structural Model

#### 3.2.1. Integrated Health Theory

The full SEM based on the IHT was overidentified (i.e., degree freedom >0) (Figure 3). In other words, the number of model parameters [{number of measured variables × (number of measured variables + 1)}/2 = (44 × 45/2) = 990] was greater than the number of free parameters estimated (*n* = 251). The total number of free parameters estimated was the sum of the number of (i) factor loadings and error terms associated with the indicators/items used for measuring the constructs, (ii) residual variances associated with the observed variables in the model other than the indicators, (iii) covariances among the constructs; and (iv) structural paths/regression coefficients.

The global model fit statistics CFI (=0.932), TLI (=0.924) and RMSEA (=0.053, 95% CI = 0.051, 0.055) values showed a good fit of the model to the observed data. The ratio of the chi-square test of model fit (χ^2^ = 4113.61) to the degrees of freedom (df = 879) (χ^2^/df = 4.68) was also less than the threshold suggested for good model fit (=5) [44,45]. However, the chi-square statistics suggested differences in the covariance matrix of the model and the observed data (χ^2^ = 4113.61, df = 829, *p* < 0.01).

#### 3.2.2. Health Belief Theory

The full SEM based on the HBT was overidentified with a higher number of model parameters identified (*n* = 741; number of measured variables in the model = 38) than the number of free parameters estimated (*n* = 239) (i.e., degree freedom = 502). The global fit statistics showed lack of fit of the model to the data (CFI = 0.625; TLI = 0.588; RMSEA = 0.111; χ^2^ = 15859.6, df = 927, *p* < 0.001).

#### 3.2.3. Theory of Planned Behavior

The full SEM based on the TPB was also overidentified with a higher number of model parameters identified (*n* = 741; number of measured variables in the model = 38) than the number of free parameters estimated (*n* = 228) (i.e., degree freedom = 513). However, the global fit statistics indicated the model does not fit the data (CFI = 0.831; TLI = 0.817; RMSEA = 0.073; χ^2^ = 7369.18, df = 916, *p* < 0.001).

### 3.3. Factors Associated with HPV Vaccination Willingness and Uptake

The full SEM drawn based on the IHT, HBM and TPB was not nested (i.e., variables were not identical). Hence, the likelihood ratio, Akaike information criterion (AIC) and Schwarz criterion (SC) values were not estimated to compare the three models. The IHT-derived framework, which fitted the data based on the global model fit statistics, was used to identify factors correlated with HPV vaccination willingness and uptake. 

The HPV vaccine uptake was significantly correlated with willingness to take the vaccine in the next 6 months (standardized regression coefficient (*β*) = 0.68, *p* < 0.001). Attitudes about the benefits (β = 0.35, *p* < 0.001) or barriers (β = −0.048, *p* = 0.023) to HPV vaccination also had a significant indirect correlation with vaccine uptake by affecting willingness to receive it (Figure 3). Nevertheless, there was no significant indirect correlation between the subjective norms related to the HPV vaccine and the vaccine uptake (β = 0.029, *p* = 0.064) (Table 3). 

Teenagers and young adults who perceived that the HPV vaccine had greater benefits were substantially more willing to get the shot (*β* = 0.52, *p* < 0.001). On the other hand, teens and young adults believing in greater perceived barriers to receiving the HPV vaccine showed a lower willingness to uptake the vaccine (β = −0.07, *p* = 0.022). On the other hand, there was no correlation between willingness to receive the HPV vaccine and subjective norms related to it (β = 0.43, *p* = 0.063).

In turn, beliefs about barriers to obtaining vaccines in general (β = −0.27, *p* < 0.001), and beliefs about susceptibility (β = 0.30, *p* = 0.005) or severity (β = 0.62, *p* < 0.001) of HPV infection and associated cancers significantly indirectly associated with willingness to uptake HPV vaccine. However, the indirect relationship between beliefs about the benefits of vaccination in general and willingness to uptake HPV vaccine through the influence of perceptions about the benefits or barriers of HPV vaccination was insignificant (β = −0.02, *p* = 0.825). The indirect relationship of normative beliefs about vaccination with a willingness to uptake HPV vaccine through the influence of subjective norms about HPV vaccine was also insignificant (β = −0.05, *p* = 0.075). 

**Figure 2 vaccines-11-01856-f002:**
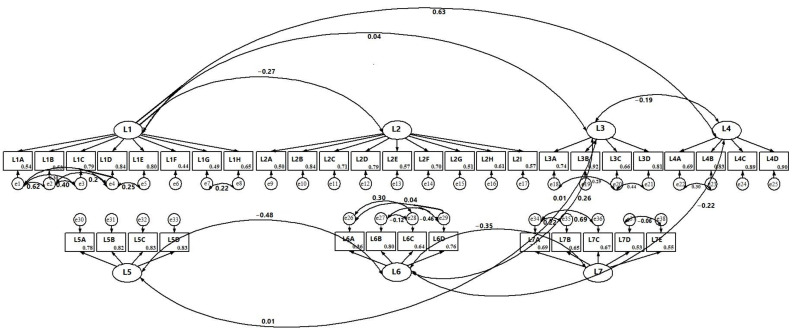
Measurement model of the constructs of integrated health theory predicting HPV vaccine uptake among teenagers and young adults in the Midwest, US, March/April 2023. L1 = benefits/facilitators of HPV vaccination; L2 = barriers to HPV vaccination; L3 = susceptibility to HPV infection and associated cancers; L4 = severity of HPV infection and associated cancers; L5 = benefits of vaccination; L6 = barriers to vaccination; L7 = awareness and knowledge about HPV infection and associated cancers and HPV vaccine. Details of names of items measuring the constructs are provided in Table 1.

**Figure 3 vaccines-11-01856-f003:**
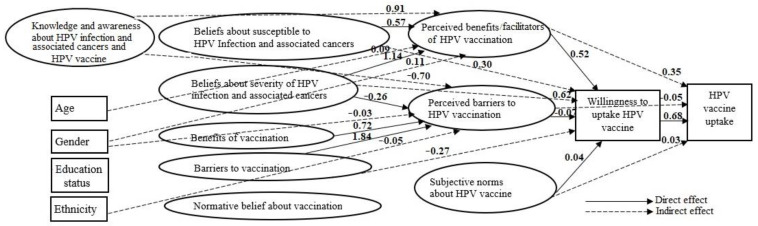
Integrated health theory-derived structural equation model explaining factors predicting HPV vaccination willingness and uptake among teenagers and young adults in Midwest, US, March/April, 2023. Parameter estimates are standardized regression coefficients. Only statistically significant coefficients are provided in this figure. Model fit statistics CFI (=0.932), TLI (=0.924) and RMSEA (=0.053, 95% CI = 0.051, 0.055).

**Table 3 vaccines-11-01856-t003:** Factors associated with HPV vaccination willingness and uptake among teenagers and young adults in Midwest, US, March/April 2023.

Exposure	Mediator	Outcome	β (95% CI)
Willingness to uptake HPV vaccine	NA	HPV vaccine uptake	0.68 (0.63, 0.72)
Perceived benefits/facilitators of HPV vaccination	NA	Willingness to uptake HPV vaccine	0.52 (0.48, 0.57)
Perceived barriers to HPV vaccination	NA	Willingness to uptake HPV vaccine	−0.07 (−0.12, −0.02)
Subjective norms of HPV vaccination	NA	Willingness to uptake HPV vaccine	0.04 (0.01, 0.08)
Perceived benefits/facilitators of HPV vaccination	Willingness to uptake HPV vaccine	HPV vaccine uptake	0.35 (0.31, 0.40)
Perceived barriers to HPV vaccination	Willingness to uptake HPV vaccine	HPV vaccine uptake	−0.05 (−0.09, −0.01)
Subjective norms of HPV vaccination	Willingness to uptake HPV vaccine	HPV vaccine uptake	0.03 (0.003, 0.06)
Beliefs about the benefits of vaccination	Perceived benefits/facilitators of HPV vaccination	Willingness to uptake HPV vaccine	0.04 (−0.09, 0.16)
Beliefs about benefits of vaccination	Perceived barriers to HPV vaccination	Willingness to uptake HPV vaccine	−0.05 (−0.11, 0.005)
Total			−0.02 (−0.16, 0.13)
Beliefs about barriers to vaccination	Perceived benefits/facilitators of HPV vaccination	Willingness to uptake HPV vaccine	−0.14 (−0.23, −0.05)
Beliefs about barriers to vaccination	Perceived barriers to HPV vaccination	Willingness to uptake HPV vaccine	−0.13 (−0.26, −0.03)
Total			−0.27 (−0.42, −0.12)
Beliefs about susceptibility to HPV infection and cervical cancer	Perceived benefits/facilitators of HPV vaccination	Willingness to uptake HPV vaccine	0.30 (0.09, 0.51)
Beliefs about susceptibility to HPV infection and cervical cancer	Perceived barriers to HPV vaccination	Willingness to uptake HPV vaccine	0.002 (−0.02, 0.02)
Total			0.30 (0.09, 0.51)
Beliefs about severity of HPV infection and cervical cancer	Perceived benefits/facilitators of HPV vaccination	Willingness to uptake HPV vaccine	0.60 (0.42, 0.78)
Beliefs about severity of HPV infection and cervical cancer	Perceived barriers to HPV vaccination	Willingness to uptake HPV vaccine	0.02 (−0.002, 0.04)
Total			0.62 (0.44, 0.79)
Normative beliefs about HPV vaccination	Subjective norms of HPV vaccination	Willingness to uptake HPV vaccine	0.02 (−0.001, 0.03)

### 3.4. Factors Associated with the Attitudes about HPV Vaccination

Beliefs about susceptibility to HPV infection and cervical cancers (β = 0.57, *p* = 0.005) and severity of HPV infection and cervical cancer (β =1.14, *p* < 0.001) were significantly positively related to the perceived benefits/facilitators of HPV vaccination (Appendix A). Beliefs about barriers to vaccination were significantly negatively related to the perceived benefits/facilitators of HPV vaccination (β = −0.27, *p* = 0.001). Beliefs about the benefits (β = 0.72, *p* = 0.012) or barriers (β = 1.84, *p* < 0.001) to vaccination were significantly positively related to perceived barriers to HPV vaccination. The belief about severity of HPV infection and cervical cancer was significantly negatively related to the perceived barriers to HPV vaccination (β = −0.26, *p* = 0.04). Knowledge and awareness about HPV infection, cancers associated with HPV infection, and HPV vaccine (β = 0.91, *p* < 0.001), higher age (β = 0.09, *p* = 0.025) and being female (β = 0.09, *p* = 0.002), were positively indirectly related to the perceived benefits/facilitators of HPV vaccination through influencing perceptions about vaccination, susceptibility and severity of HPV infection and associated cancers.

An increase in the knowledge and awareness about HPV infection, cervical cancer and HPV vaccine was associated with an increase in the beliefs about the benefits of vaccination (β = 0.697, *p* < 0.001) and severity of HPV infection and associated cancers (β =0.73, *p* < 0.001), but negatively associated with the beliefs about barriers to vaccination (β = −0.56, *p* < 0.001), and susceptibility to HPV infection and associated cancers (β = −0.21, *p* = 0.02). Higher age teenagers and young adults were more likely to believe they were at risk of HPV infection and associated cancers (β = 0.10, *p* = 0.090). Females were more likely to believe that HPV infection and associated cancers are severe (β = 0.10, *p* < 0.001) and that vaccination is beneficial (β = 0.07, *p* = 0.028).

## 4. Discussion

This study applied SEM to identify a health theory that best explains the nature of the relationship between the factors associated with HPV vaccine uptake and used the information to determine predictors of HPV vaccine uptake by teenagers and young adults in the US. The IHT was better at explaining predictors of HPV vaccine uptake than the HBT and TPB. While willingness to uptake the HPV vaccine directly positively predicted vaccine uptake, attitudes about the vaccine indirectly predicted the uptake. In turn, beliefs towards vaccination barriers and the likelihood and severity of HPV infection and related cancers indirectly correlated with willingness to receive the vaccine.

An acceptable fit of the IHT-derived SEM framework to explain the relationships of determinants of HPV vaccine uptake data in teenagers and young adults of the US population confirmed a similar finding among parents in the Indian population [29]. The IHT has all the constructs of the TPB and HBT plus additional elements [20,21] and posits background variables and knowledge about HPV infection and related cancers and the vaccine as predictors of beliefs about these conditions [20,21]. In addition, in contrast to the HBT, the IHT suggests willingness to uptake the vaccine as a direct predictor of the vaccine [20,21]. These features could have better allowed IHT to explain the complex and overlapping nature of the relationship of the factors affecting HPV vaccine uptake.

Teenagers and young adults who were willing to obtain an HPV vaccination reported a high rate of uptake of the vaccine. Several studies also showed an association between willingness to uptake the HPV vaccine and increased actual vaccine uptake in different US populations [18,25,26,32,33]. Perceived benefits/facilitators to HPV vaccination, determined by evaluating the favorable perception of the vaccine or factors that promote vaccination, predicted increased uptake by increasing willingness to uptake it. On the other hand, negative attitudes regarding the HPV vaccine predicted a decreased uptake of the vaccine by reducing willingness to receive it. In addition, teenagers and young adults who believe society expects them to be vaccinated are interested in it. Studies among teenagers in another region of US also reported a positive relationship between perceptions about the benefits of the HPV vaccine and willingness to uptake the HPV vaccine [18,25,26,32,33], and perceived barriers to vaccine uptake with a decreased vaccine uptake [18,25,26,32,33].

In turn, teenagers and young adults were more likely to perceive that the HPV vaccine is helpful if they believed that they were at risk of HPV infection and related cancers, that these conditions are serious, and that vaccination is beneficial. On the other hand, teenagers and young adults who believed they were less susceptible to HPV infection and associated cancers and that HPV infection and associated cancers are less severe and vaccination is less beneficial were less likely to perceive that HPV vaccine is beneficial. These results concurred with earlier studies [18,25,26,32,33]. As the HPV vaccination prevents HPV infection and associated cancers, individuals who believe that they are at risk of contracting HPV infection and related cancers and the severity of these diseases will likely have positive perceptions about the vaccine. Similarly, positive beliefs about vaccination could positively influence the attitude toward the HPV vaccine.

The current findings inform an IHT-based framework to study factors that affect a decision to uptake the HPV vaccine and understand how these factors interact to influence HPV vaccination. This study also identified reliable items to measure the constructs of advanced health theories suggested for studying health behavior including IHT that predict HPV vaccination. This will advance the IHT as an explanatory framework for HPV vaccination research and provide insight into theory-based behavior change interventions that could help reduce HPV vaccine hesitancy and increase acceptance of the HPV vaccine in the Midwest and other regions in the US. This, in turn, contributes to the development of societally acceptable and relevant interventions to effectively increase HPV vaccination among teenagers and young adults in the Midwest region in the US. After adjusting to the population’s culture and other relevant characteristics, the current study findings can guide interventions to promote HPV vaccination in other states of the US. Policy measures to promote HPV vaccination and reduce associated cancer rates in the Midwest population may focus on changing negative attitudes and norms associated with the HPV vaccine, HPV infection and associated cancers among teenagers and young adults. The policies may involve educational initiatives to dispel myths about the risk of HPV infection and related cancers as well as injection pain phobia and objections from religious authorities and friends who are still too young to be vaccinated within the target population. To ensure an increased HPV vaccination rate in the target population, the education programs could also be broadened to include parents and religious authorities who may have influence on the decision-making of teenagers and young adults to uptake the HPV vaccine. In order to reduce the false perceptions of the teenagers and young adults about the vaccine and HPV infection, the content of the education program may include the ways in which HPV infections spread and result in diseases, the knowledge that vaccination against HPV can prevent cancers linked to the virus and how to make an appointment to get the HPV vaccine.

This study had a high response rate and a sample size that was substantial. Using SEM, a solid theory-based analysis was carried out. The final SEM framework used to identify predictors of HPV vaccine uptake in the population acceptably fitted the data. Nonetheless, the following limitations should be considered when interpreting the results. The chi-square value indicated a difference in the covariance matrix between the IHT-generated framework and the present data, even if the CFI, TLI, and RMSEA values verified that the IHT-derived framework fit the data. However, the chi-square test will likely be significant as the sample size increases (threshold of 200). In addition, the ratio of the chi-square test of model fit (χ^2^ = 4113.61) to the degrees of freedom (df = 879) (χ^2^/df = 4.68) was also less than the threshold suggested for good model fit by some scholars (=5) [44,45]. Hence, the validity of the fit of the current model to explain the data is justifiable. In addition, the cross-sectional aspect of this study should be taken into consideration when interpreting the mediating impact. Moreover, there could be information bias in responses. There could have been a gap, extending to years, between the time when the teenagers and young adults were vaccinated and the data collection period. Hence, the current study participants may not correctly remember their HPV vaccination status. Indeed, over 35% of the participants reported that they were unsure about their HPV vaccination status. Furthermore, the ethnic composition of the sample does not fully represent the Midwest population. While the ethnic composition in the Midwest is approximately 73% White, 10% Black, 3% Asian, 9% Hispanic, 4% two ethnic groups and 0% Native or Islander [46], the composition in the current sample was 58.1% white, 24.0% Black or African, 5.3% American Indian or Alaska natives, and 3.1% Asian. Finally, the survey should also target parents as they will decide on vaccine uptake for their younger teens.

## 5. Conclusions

IHT can be appropriate to examine predictors of HPV vaccine uptake in the teenager and young adult population in the US, particularly in the Midwest. Future studies studying predictors of HPV vaccine uptake could apply IHT to guide the survey design, implementation, and analysis. Negative attitudes and norms associated with the HPV vaccine, misbeliefs about HPV infection and related cancers and vaccination might have caused the reduced complete HPV vaccine uptake rate in the Midwest population in the US. Educational interventions to improve HPV vaccine uptake in the Midwest population could focus on creating awareness about the HPV vaccine, HPV infection and related cancers and vaccination in the target population.

## Figures and Tables

**Table 1 vaccines-11-01856-t001:** Constructs of the health theories and the corresponding items used for measuring them along with their responses/scores.

Constructs	Item Label	Item	Responses/Scores
Benefits/facilitators of HPV vaccination (L1)	L1A	My doctor recommends me to receive HPV vaccine	0. Strongly disagree,
			1. Disagree,
			2. Neutral,
			3. Agree,
			4. Strongly agree
	L1B	My family member recommends/supports me to receive HPV vaccine	
	L1C	I believe HPV vaccine is beneficial to my health	
	L1D	I believe that HPV vaccine is safe	
	L1E	I believe that HPV vaccine is effective	
	L1F	I believe that I will become sexually active	
	L1G	I believe that HPV infection can cause cervical cancer	
	L1H	I believe that HPV vaccine will prevent cervical, oropharyngeal, vaginal, vulvar, penile, anal, and rectal cancers for self and others	
Barriers to HPV vaccination (L2)	L2A	HPV vaccine is too expensive	0. Strongly disagree,
			1. Disagree,
			2. Neutral,
			3. Agree,
			4. Strongly agree
	L2B	My religious belief goes against me getting HPV vaccine	
	L2C	Objections in getting HPV vaccine from religious authorities will prevent me from getting the vaccine	
	L2D	Friends who disapprove of getting HPV vaccine will prevent me from getting the vaccine	
	L2E	I am afraid that HPV vaccine injection may cause pain	
	L2F	I will not have time to get HPV vaccine	
	L2G	I do not know how to make an appointment to get HPV vaccine	
	L2H	My health insurance does not cover the HPV vaccine	
	L2I	I am too young for getting vaccination	
Susceptibility to HPV infection and associated cancers (L3)	L3A	I believe I am at risk of getting HPV infection	0. Strongly disagree,
			1. Disagree,
			2. Neutral,
			3. Agree,
			4. Strongly agree
	L3B	I will likely contract HPV infection	
	L3C	I am at risk of getting cervical, oropharyngeal, vaginal or penile, anal or rectal cancer	
	L3D	I will likely contract cervical, oropharyngeal, vaginal, or penile or rectal cancer	
Severity of HPV infection and associated cancers (L4)	L4A	I believe that HPV infection is severe	0. Strongly disagree,
			1. Disagree,
			2. Neutral,
			3. Agree,
			4. Strongly agree
	L4B	I believe that HPV infection is serious.	
	L4C	I believe that cervical, oropharyngeal, vaginal, vulvar, penile, anal, and rectal cancers are severe	
	L4D	I believe that cervical, oropharyngeal, vaginal, vulvar, penile, anal, and rectal cancers are serious	
Benefits of vaccination (L5)	L5A	Vaccines are effective in preventing disease	0. Strongly disagree,
			1. Disagree,
			2. Neutral,
			3. Agree,
			4. Strongly agree
	L5B	It is very important that I receive all the necessary routine vaccines	
	L5C	Vaccine is one way that I can ensure good health	
	L5D	I have a responsibility to get vaccinated for the protection of others	
Barriers to vaccination (L6)	L6A	I am concerned about vaccine side effects	0. Strongly disagree,
			1. Disagree,
			2. Neutral,
			3. Agree,
			4. Strongly agree
	L6B	It is better to get the disease and get protected naturally	
	L6C	There are too many vaccines available to take	
	L6D	I have a negative experience with vaccination	
Awareness and knowledge about HPV infection and associated cancers and HPV vaccine (L7)	L7A	Have you ever heard about the HPV infection?	1. Yes, 2 = No
	L7B	Have you ever heard about cervical cancer, anal, penial, vaginal, vulva, oropharynx?	1. Yes, 0 = No
	L7C	Have you ever heard about the HPV vaccine?	1. Yes, 0 = No
	L7D	HPV infection can cause cervical, oropharyngeal, vaginal, vulvar, penile, anal, and rectal cancers	1. True, 0 = False
	L7E	HPV vaccine can prevent cervical, oropharyngeal, vaginal, vulvar, penile, anal, and rectal cancers	1. True, 0 = False

**Table 2 vaccines-11-01856-t002:** Sociodemographic characteristics of the study sample.

Variables	Categories	Illinois (*n* = 259)	Ohio (*n* = 243)	Michigan (*n* = 187)	Indiana (*n* = 126)	Missouri (*n*= 117)	Wisconsin (*n* = 97)	Minnesota (*n* = 91)	Kansas (*n* = 58)	Iowa (*n* = 53)	Nebraska (*n*= 42)	South Dakota (*n* = 17)	North Dakota (*n* = 16)	Total (*n* = 1306)
Age in years	13–17	42.9	40.3	41.2	38.1	47.9	44.3	52.75	53.5	49.1	59.5	41.18	56.3	44.3
	18–22	40.9	43.2	42.8	48.4	38.5	41.2	28.6	36.2	43.4	23.8	35.3	37.5	40.5
	23–26	16.2	16.5	16.0	13.5	13.7	14.4	18.7	10.3	7.6	16.7	23.5	6.3	14.4
Gender	Male	45.2	55.1	51.9	50.8	53.0	55.67	47.3	43.1	45.3	54.8	47.1	68.8	50.7
	Female	51.7	42.4	40.1	43.7	42.7	43.3	48.4	53.4	45.3	40.4	41.2	31.3	45.0
	Other	3.1	2.5	8.0	5.6	4.3	1.0	4.4	3.5	9.4	4.8	11.8	0.0	4.3
Educational attainment	Less than high school	34.4	37.4	36.9	35.7	42.7	46.4	41.8	46.6	50.9	47.6	64.7	43.8	39.7
	High school graduate or GED	35.5	34.1	33.2	38.1	35.0	29.9	37.4	25.9	30.2	28.6	11.8	31.3	33.6
	Some college	20.5	22.63	20.3	18.25	17.1	19.6	13.2	22.4	17.0	14.3	23.5	18.8	19.5
	Graduate school	9.7	5.8	9.6	7.9	5.1	4.1	7.7	5.2	1.9	9.5	0.0	6.3	7.1
Race	White	48.3	60.9	53.5	54.8	62.6	68.0	62.6	65.5	71.7	69.1	64.7	62.5	58.1
	Black or African American	31.3	26.3	29.4	28.6	16.5	9.3	16.5	12.1	5.7	9.5	11.8	18.8	24.0
	American Indian or Alaska natives	1.9	1.2	2.1	2.4	3.5	5.15	3.3	5.2	1.9	4.76	23.5	12.5	5.3
	Asian	5.41	4.9	5.4	5.6	8.8	7.2	8.8	3.4	7.5	2.4	0.0	6.3	3.1
	Native Hawaiian or Pacific Islander	0.0	1.23	1.6	0.0	0.0	0.0	0.0	0.0	1.9	0.0	0.0	0.0	0.5
	Other	13.1	5.4	8.0	8.7	8.8	10.3	8.79	13.79	11.32	14.29	0.0	0.0	8.9
Ethnicity	Hispanic or Latino	23.17	14.0	10.2	16.7	11.97	21.7	19.8	31.0	9.4	21.4	5.9	18.8	17.1
	Non-Hispanic or non-Latino	76.8	86.0	89.8	83.3	88.0	78.4	80.2	69.0	90.6	78.6	94.1	81.3	82.9
HPV vaccination status	Fully vaccinated (3 doses)	15.1	6.3	13.4	15.1	10.3	14.4	15.4	15.5	11.3	9.52	0.0	6.3	13.6
	Fully vaccinated (2 doses)	16.2	18.8	16.1	9.5	16.2	15.5	18.7	13.8	7.6	23.8	29.4	18.8	16.8
	Partially vaccinated (one does)	10.0	12.5	13.4	8.7	5.98	7.2	6.6	6.9	9.4	9.52	5.9	12.50	9.5
	Unvaccinated	22.4	25.0	21.9	26.2	29.9	25.8	27.5	20.7	24.5	23.8	23.5	25.0	24.8
	Do not know	36.3	37.5	35.3	40.5	37.6	37.1	31.9	43.1	47.2	33.3	41.2	37.5	37.1

Values in the table are in percent.

## Data Availability

The data presented in this study are available on request from the corresponding author. The data are not publicly available due to privacy/ethical issues.

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
