# Peer review of "Evaluation of Theoretical Frameworks to Detect Correlates of HPV Vaccination in the Midwest, US, Using Structural Equation Modeling"

_vaccines, 2023, doi:10.3390/vaccines11121856_

Round 1
Reviewer 1 Report
Comments and Suggestions for Authors
The authors address an important public health topic in identifying theoretically based factors that directly or indirectly affect teenagers' and young adults’ HPV vaccine uptake. The strengths of the study of the manuscript include:
1. This is a well-written paper with sufficient background information to support the need for this work.
2. The authors provided details about the application of the theoretical approach which provides a template for easy replication of the study in different settings.
3. The approach produced impressive results, that show predictors of HPV vaccination uptake among teenagers and young adults.
4. The future directions the authors provided are very critical.
However, I have a few concerns:
1. In Table 1, I wonder why “My doctor recommends me to receive HPV vaccine” and “My family member recommends/supports me to receive HPV vaccine” are considered as benefits of HPV vaccine items. Believe these are subjective norm items instead.
2. Use Healthy People 2030 indicators instead of the Healthy People 2020 goal of 80% in females.
3. In section 2.4, the authors stated, “Panelists were invited to participate and opted in by activating a survey link directing them to the study consent and survey instrument.” This is not clear. Are they the panel of experts or who are they? Need to describe them and their function. Were they used for content and face validity or what did they do?
4. In section 2.4, the authors stated, “Only those who assented completed the survey” The authors need to clarify this statement. The study participants included teenagers and young adults. For the teenagers, they will need parental assent and for the young adults, they will need informed consent. The authors need to describe how parental assent and participants' informed consent were obtained.
5. In section 2.5 the authors stated:
“First, a Cronbach's alpha coefficient calculated to evaluate the reliability/internal consistency of the 184 items in measuring the constructs of the health theories (α ≥ 0.7 acceptable).” Acceptable per what standard? Providing reference(s) will suffice the standard criteria.
“Then, confirmatory factor analysis was used to assess the validity of the items (factor loading ≥ 0.4 acceptable), the association between latent factors, and to check the fit of the measurement model to the data.” Same issue. Provide reference for the 0.4 factor loading considered acceptable.
6. Check the spelling of Cronbach alpha (authors spelt “Chronbach”)
7. Recruitment methods: The authors projected a sample size of 1,306 and I am curious about how they were able to obtain 100% participation rate? Just curious. “A total of 1,306 teenagers and young adults (age range: 13-26 years) participated in 203 this study.”
Providing or describing how participants were recruited and the survey was administered will demonstrate the rigorousness of the study approach and the credibility of the findings.
Comments on the Quality of English LanguageMinor typos
Author Response
Reviewer 1 The authors address an important public health topic in identifying theoretically based factors that directly or indirectly affect teenagers' and young adults' HPV vaccine uptake. The strengths of the study of the manuscript include: 1. This is a well-written paper with sufficient background information to support the need for this work. 2. The authors provided details about the application of the theoretical approach which provides a template for easy replication of the study in different settings. 3. The approach produced impressive results, that show predictors of HPV vaccination uptake among teenagers and young adults. 4. The future directions the authors provided are very critical. However, I have a few concerns: 1. In Table 1, I wonder why "My doctor recommends me to receive HPV vaccine" and "My family member recommends/supports me to receive HPV vaccine" are considered as benefits of HPV vaccine items. Believe these are subjective norm items instead. Response: thank you for catching this. The construct' benefits of HPV vaccine' included items that were used to assess perceived facilitators of HPV vaccine (e.g. "My doctor recommends me to receive HPV vaccine" and "My family member recommends/supports me to receive HPV vaccine"). We have revised the text/construct "benefits of HPV vaccine" as "benefits/facilitators of HPV vaccine" throughout the manuscript. 2. Use Healthy People 2030 indicators instead of the Healthy People 2020 goal of 80% in females. Response: We have updated the information in the background based on the Healthy People 2030 indicators. The revised text reads, "In 2021, the complete HPV vaccination rate (2 or 3 doses) among adolescents aged 13 to 15 was only 58.5%, far lower than the Healthy People 2030 goal of 80%." (please see page 1, line 41-43). 3. In section 2.4, the authors stated, "Panelists were invited to participate and opted in by activating a survey link directing them to the study consent and survey instrument." This is not clear. Are they the panel of experts or who are they? Need to describe them and their function. Were they used for content and face validity or what did they do? Response: We have added a text that describes the panelists in Qualtrics. The added text reads " Qualtrics has over 90 million online samples or market research panelists used for corporate and academic research. The panelists completed a standardized set of questions to create their profile in Qualtrics. The profiles were used to choose eligible respondents at random for surveys. For this study, Qualtrics online samples used a combination of actively managed, double-opt-in market research panels to recruit participants. The study participants were existing pools of research panel samples of the Qualtrics with ages 13 to 26 years living in the Midwest area in the U.S. …….. A random sample of research panels in the Qualtrics with ages 13 to 26 years living in the Midwest area in the U.S., who have agreed to be contacted for research studies, received an invitation email to participate in the online survey. Individuals were invited to participate in the survey if they were preregistered for Qualtrics Panels and completed a baseline proprietary survey." (please see section 2.1, page # 3, lines 112 to 133). 4. In section 2.4, the authors stated, "Only those who assented completed the survey" The authors need to clarify this statement. The study participants included teenagers and young adults. For the teenagers, they will need parental assent and for the young adults, they will need informed consent. The authors need to describe how parental assent and participants' informed consent were obtained. Response: We acknowledge your comment and have revised section 2.4 to explain why parental assent was not obtained and how consent was obtained from the participants. The revised ethical consideration text in section 2.4 reads "The study obtained ethical approval from the Institutional Review Board (IRB) of the University of Nebraska Medical Center (IRB # 0696-22-EP). As the study has a very minimal risk on the participants, who were existing pools of research panel lists of a data processor company―Qualtrics, the IRB waived parental or guardian consent/permission for teenagers. Panelists were invited to participate through email and opted in by activating a link that directed them to the study consent page and survey instrument. The participants assented by simply checking a box after reading the consent sheet with details about the study's purpose, confidentiality, risks, and benefits. Only those who assented completed the survey." (please see section 2.4, lines 189 to 197). 5. In section 2.5, the authors stated: "First, a Cronbach's alpha coefficient was calculated to evaluate the reliability/internal consistency of the 184 items in measuring the constructs of the health theories (α ≥ 0.7 acceptable)." Acceptable per what standard? Providing reference(s) will suffice the standard criteria. "Then, confirmatory factor analysis was used to assess the validity of the items (factor loading ≥ 0.4 acceptable), the association between latent factors, and to check the fit of the measurement model to the data." Same issue. Provide a reference for the 0.4 factor loading considered acceptable. Response: Thank you for catching this. We have provided references for acceptable values of Cronbach (≥0.7) and factor loadings (≥0.4) (please see pages 8, lines 196 & 198). Cronbach alpha A. Tavakol M, Dennick R. Making sense of Cronbach's alpha. Int J Med Educ. 2011;2:53-55. B. Bland J, Altman D. Statistics notes: Cronbach's alpha. BMJ. 1997;314:275. 10.1136/bmj.314.7080.572 Reference for factor loading ≥ 0.4 acceptable Hair J.F., Anderson R.E., Babin B.J., Black W.C. Multivariate Data Analysis: A Global Perspective. 7th ed. Prentice Hall; Upper Saddle River, NJ, USA: 2009. 6. Check the spelling of Cronbach alpha (authors spelled "Chronbach") Response: We have edited 'Chronbach' as 'Cronbach. (Please read section "2.5" line 201 and section 3.1.1) 7. Recruitment methods: The authors projected a sample size of 1,306 and I am curious about how they were able to obtain 100% participation rate? Just curious. "A total of 1,306 teenagers and young adults (age range: 13-26 years) participated in 203 this study." Providing or describing how participants were recruited and the survey was administered will demonstrate the rigorousness of the study approach and the credibility of the findings. Response: thank you for catching this. We have requested Qualtrics to complete the survey by 1288 panels (i.e., the estimated sample size). Fortunately, Qualtrics provided us with data for 1306 individuals. I believe they assumed all the individuals contacted may not agree to participate or complete the survey. So, they invited over 1288 individuals to participate in the survey and found 1306 individuals who completed it. They have likely invited over 1306 individuals to participate. However, we couldn’t get exact information on the number of panels who were invited to participate. So, we have edited the text regarding the study participation rate in section 2.3. The revised text reads ‘Thus, we needed a minimum of 1,288 participants to adequately test the validity of the conceptual frameworks based on the IHT, TPB, H.B.T. Fortunately, Qualtrics collected and provided data for 1,306 individuals.” (please see section 2.3, lines 184 to 186). In addition, we have provided details on the study participants' recruitment procedure in the revised manuscript. The revised text reads, “A random sample of research panels in the Qualtrics with ages 13 to 26 years living in the Midwest area in the U.S., who have agreed to be contacted for research studies, received an invitation email to participate in the online survey. Individuals were invited to participate in the survey if they were preregistered for Qualtrics Panels and completed a baseline proprietary survey. The typical survey invitation through email was generally very simple and generic and, informed the estimated completion time and incentives, and contained a hyperlink that directed the respondent to the study consent page and the survey instrument, which was completed through Qualtrics. Survey invitations through email did not include specific details about the questionnaire contents and were instead kept very general to avoid self-selection bias. Participants were informed that the survey was for research purposes, and confidentiality, risks, and benefits were described at the beginning of the survey. After they assent, participants were directed to the questionnaire; after completing it, they were compensated. Qualtrics staff monitored responses.”

Reviewer 2 Report
Comments and Suggestions for Authors
The authors write, “In 19 turn, beliefs about susceptibility (p=0.05) and severity (p<0.001) of HPV infection and associated 20 cancers and barriers to vaccination in general (p<0.001) indirectly predicted willingness to uptake 21 the vaccine”. Presenting the p-value to be exactly equal to 0.05 is problematic because the rule is. The null hypothesis is rejected if p < 0.05, and the null hypothesis is not rejected if p > 0.05. If the rounded value of the p-value displays as 0.05 (when rounded to 2 decimal places), the authors should express the p-value by using more decimal places so that one can see whether the p-value is, for example, p = 0.0499 (< 0.05) or p = 0.0511 (> 0.05).
The authors write “factor loading ≥ 0.4 acceptable”. How do the authors justify that it’s acceptable? Lambert and Newman (2023) wrote, “We note a common rule of thumb that standardized factor loadings should be >= 0.4, consistent with a similar heuristic in the context of exploratory factor analysis (Ford, MacCallum, & Tait, 1986). This cutoff is arbitrary, but is also a fairly low standard. Factor loadings>= 0.4 mean that the latent factor accounts for at least 16% of variance (0.4 squared) in the measure (it also implies that interitem correlations are at least r = 0.16). All else equal, larger loadings are generally better”.
Lambert, L. S., & Newman, D. A. (2023). Construct development and validation in three practical steps: Recommendations for reviewers, editors, and authors. Organizational Research Methods, 26(4), 574-607. https://doi.org/10.1177/10944281221115374
Not all the layers of the research onion (Saunders et al., 2009) have been considered:
First layer (and the outer-most layer): Philosophy (i.e. positivism, interpretivism, pragmatism etc.)
Second layer: Approach to theory development (i.e. deduction, induction etc.)
Third layer: Methodological choice (i.e. quan, qual, mixed-methods etc.).
Fourth layer: Strategies (i.e. case study, action research, etc.)
Fifth layer: Time horizon (i.e. cross-sectional or longitudinal)
Sixth (and the inner-most layer): Techniques and procedures (i.e. the data collection and data analysis).
Reference:
Saunders, M., Lewis, P., & Thornhill, A. (2009). Research methods for business students (5th ed.). Pearson Education Limited.
Author Response
Reviewer 2
Comments and suggestions for Authors
- The authors write, "In 19 turn, beliefs about susceptibility (p=0.05) and severity (p<0.001) of HPV infection and associated 20 cancers and barriers to vaccination in general (p<0.001) indirectly predicted willingness to uptake 21 the vaccine". Presenting the p-value to be exactly equal to 0.05 is problematic because the rule is. The null hypothesis is rejected if p < 0.05, and the null hypothesis is not rejected if p > 0.05. If the rounded value of the p-value displays as 0.05 (when rounded to 2 decimal places), the authors should express the p-value by using more decimal places so that one can see whether the p-value is, for example, p = 0.0499 (< 0.05) or p = 0.0511 (> 0.05).
Response: thank you for catching this typo. We have corrected p = 0.05 as p = 0.005. Please see pages 1 (line 20), 16 (line 32), and 17 (line 45)
- The authors write "factor loading ≥ 0.4 acceptable". How do the authors justify that it's acceptable? Lambert and Newman (2023) wrote, "We note a common rule of thumb that standardized factor loadings should be >= 0.4, consistent with a similar heuristic in the context of exploratory factor analysis (Ford, MacCallum, & Tait, 1986). This cutoff is arbitrary, but is also a fairly low standard. Factor loadings>= 0.4 mean that the latent factor accounts for at least 16% of variance (0.4 squared) in the measure (it also implies that interitem correlations are at least r = 0.16). All else equal, larger loadings are generally better".
Lambert, L. S., & Newman, D. A. (2023). Construct development and validation in three practical steps: Recommendations for reviewers, editors, and authors. Organizational Research Methods, 26(4), 574-607. https://doi.org/10.1177/10944281221115374.
Response: We set the cutoff at 0.4 for the acceptable factor loading values following the rule of thumb. Fortunately, except for one (factor loading=0.4), all the items loaded significantly with a factor loading ≥0.5.
- Not all the layers of the research onion (Saunders et al., 2009) have been considered: First layer (and the outer-most layer): Philosophy (i.e. positivism, interpretivism, pragmatism etc.) Second layer: Approach to theory development (i.e. deduction, induction etc.) Third layer: Methodological choice (i.e. quan, qual, mixed-methods etc.). Fourth layer: Strategies (i.e. case study, action research, etc.) Fifth layer: Time horizon (i.e. cross-sectional or longitudinal) Sixth (and the inner-most layer): Techniques and procedures (i.e. the data collection and data analysis).
Reference:
Saunders, M., Lewis, P., & Thornhill, A. (2009). Research methods for business students (5th ed.). Pearson Education Limited.
Response: We are unsure if this comment relates to our paper.

Reviewer 3 Report
Comments and Suggestions for Authors
Degarege et al. have conducted a survey on knowledge, attitude and beliefs on HPV vaccine and they have correlated it to HPV vaccine uptake, using a sample of 1306 teenagers and young adults in the US Midwest. They have tested different models and concluded that the Integrated Health Theory model fits best with the data. The results show that willingness to be vaccinated, perception of HPV infections severity and susceptibility can positively predict the vaccine uptake, while perceive barriers to vaccination, among other factors, negatively predict vaccine uptake.
The authors conclude that IHT analysis has identified some positive and negative predictors of vaccine uptake, that could be used to focus public health interventions aimed at increase the coverage of HPV vaccination.
This study uses a reasonably large samples and the analysis is rigorous. The outcomes are as expected and in agreement with some previous literature, but this model could be generalised to further investigation on HPV vaccine hesitancy.
I only have some minor comments:
Line 38. “has slow and stalled in some years”. This sentence does not seem clear.
Lines 171-173. A 100% participation is astonishing, as normally just a small fraction of contacted individuals do participate. The authors state above that the participant were selected from an existing pool. Could the authors describe this pool in more detail? Could this sample be biased in term of education or positive attitudes or socioeconomic background? Since this a pool of respondent that was used before, perhaps these data are available.
Table 2 – Please, specify in the title and in the text that this table refers to the study samples, not the teenagers and young adults in the US Midwest in general.
Author Response
Reviewer 3
Comments and Suggestions for Authors
Degarege et al. have conducted a survey on knowledge, attitude and beliefs on HPV vaccine and they have correlated it to HPV vaccine uptake, using a sample of 1306 teenagers and young adults in the US Midwest. They have tested different models and concluded that the Integrated Health Theory model fits best with the data. The results show that willingness to be vaccinated, perception of HPV infections severity and susceptibility can positively predict the vaccine uptake, while perceive barriers to vaccination, among other factors, negatively predict vaccine uptake.
The authors conclude that IHT analysis has identified some positive and negative predictors of vaccine uptake, that could be used to focus public health interventions aimed at increase the coverage of HPV vaccination. This study uses a reasonably large sample and the analysis is rigorous. The outcomes are as expected and in agreement with some previous literature, but this model could be generalized to further investigation on HPV vaccine hesitancy.
I only have some minor comments:
- Line 38. "has slow and stalled in some years". This sentence does not seem clear.
Response: We have revised the sentence. The revised text reads “the increase in the rate of uptake among eligible individuals (ages 13 to 26 years) in the US, particularly in some states of the Midwest region, is slow, lowering its potential public health impact.” (Please see page 1, line 38).
- Lines 171-173. A 100% participation is astonishing, as normally just a small fraction of contacted individuals do participate. The authors state above that the participant were selected from an existing pool. Could the authors describe this pool in more detail? Could this sample be biased in term of education or positive attitudes or socioeconomic background? Since this a pool of respondent that was used before, perhaps these data are available.
Response: thank you for catching this.
- Participation rate: We have requested Qualtrics to complete the survey by 1288 panels (i.e., the estimated sample size). Fortunately, Qualtrics provided us with data for 1306 individuals. I believe they assumed all the individuals contacted may not agree to participate or complete the survey. So, they invited over 1288 individuals to participate in the survey and found 1306 individuals who completed the survey. It’s very likely that they have invited over 1306 individuals to participate. However, we couldn’t get exact information on the number of panels who were invited to participate. So, we have edited the text regarding the study participation rate in section 2.3. The revised text reads ‘Thus, we needed a minimum of 1,288 participants to adequately test the validity of the conceptual frameworks based on the IHT, TPB, H.B.T. Fortunately, Qualtrics collected and provided data for 1,306 individuals.” (section 2.3, lines 184 to 186).
- Qualtrics panels: In addition, we have provided details on the Qualtrics panels and study participants' recruitment procedure in the revised manuscript. The revised text reads
“Qualtrics has over 90 million online samples or market research panelists used for corporate and academic research. The panelists completed a standardized set of questions to create their profile in Qualtrics. The profiles were used to choose eligible respondents at random for surveys. For this study, Qualtrics online samples used a combination of actively managed, double-opt-in market research panels to recruit participants. The study participants were existing pools of research panel samples of the Qualtrics with ages 13 to 26 years living in the Midwest area in the U.S. (i.e., Illinois, Ohio, Michigan, Indiana, Missouri, Wisconsin, Minnesota, Kansas, Iowa, Nebraska, South Dakota, North Dakota) and have agreed to be contacted for research studies. Sampling was done based on gender (Female ~50% and Male ~50%) and age group (Age in years 13-17 ~45%; Age in years 18-22 ~40%; Age in years 23-26 ~15%) from the 12 states. Qualifying screening questions presented at the beginning of the survey were used to select eligible participants. After screening eligible participants, demographic screening questions were used to ensure the demographic distribution of the sample was representative of the Midwest population. The inclusion of participants was completed when quotas for each demographic group were reached.” (Please see section 2.1, page 3 line 112 to 127).
- Could this sample be biased in terms of education or positive attitudes or socioeconomic background?
Due to the nature of the sampling procedure, we believe demography, education, and attitude-related biases are less likely in this study.
Demography: the revised text reads “Sampling was done based on gender (Female ~50% and Male ~50%) and age group (Age in years 13-17 ~45%; Age in years 18-22 ~40%; Age in years 23-26 ~15%) from the 12 states. Qualifying screening questions presented at the beginning of the survey were used to select eligible participants. After screening eligible participants, demographic screening questions were used to ensure the demographic distribution of the sample was representative of the Midwest population. The inclusion of participants was completed when quotas for each demographic group were reached.” (section 2.1; page 11, lines 12-127) Education: See Table 2
Attitude : the revised text reads, “Survey invitations through email did not include specific details about the questionnaire contents and were instead kept very general to avoid self-selection bias” (section 2.1; page 3; line 136-138)
- Table 2 – Please, specify in the title and in the text that this table refers to the study samples, not the teenagers and young adults in the US Midwest in general.
Response: We have revised the title for Table 2. It reads 'Sociodemographic characteristics of the study sample".
In the text, it reads “A total of 1,306 teenagers and young adults (age range: 13-26 years) participated in this study (Table 2)”.
